# Overcoming Bureaucratic Resistance: An Analysis of Barriers to Climate Change Adaptation in South Africa

Nomfundo Patricia Sibiya [1,*], Dillip Kumar Das [2], Coleen Vogel [3], Sonwabo Perez Mazinyo [4], Leocadia Zhou [4], Mukalazi Ahmed Kalumba [4], Mikateko Sithole [5], Richard Kwame Adom [1] and Mulala Danny Simatele [1,3]

1   School of Geography, Archaeology and Environmental Studies, Faculty of Science, University of the Witwatersrand, Johannesburg 2050, South Africa; 1645009@students.wits.ac.za (R.K.A.); mulala.simatele@wits.ac.za (M.D.S.)
2   School of Engineering, College of Agriculture, Engineering and Science, University of KwaZulu-Natal, Durban 4000, South Africa; dasd@ukzn.ac.za
3   Global Change Institute, University of the Witwatersrand, Johannesburg 2050, South Africa; coleen.vogel@wits.ac.za
4   Department of Geography and Environmental Science, University of Fort Hare, Alice 5700, South Africa; smazinyo@ufh.ac.za (S.P.M.); lzhou@ufh.ac.za (L.Z.); akalumba@ufh.ac.za (M.A.K.)
5   Department of Forestry, Fisheries, and the Environment, Pretoria 0001, South Africa; mfsithole@dffe.gov.za
*   Correspondence: 573380@students.wits.ac.za

**Abstract:** Climate change is already a reality, and it is affecting the lives and livelihoods of many people globally. Many scientists argue that adaptation is, therefore, necessary to address the impact of climate change on life-supporting systems. Climate change adaptation, however, is a complex process that involves transformations implemented through governance at multiple levels. In this paper, the barriers to climate change adaptation in South Africa are presented and analysed. Semi-structured, in-depth interviews were conducted telephonically and online via Microsoft Teams with 13 government officials working at the Department of Forestry, Fisheries, and the Environment; the KwaZulu-Natal Department of Economic Development and Environmental Affairs; and the uMkhanyakude District Municipality. The findings suggest that the barriers to climate change adaptation in South Africa include inadequate financial resources, a lack of human capacity at the provincial and local levels, limited political will at the local level, limited understanding of climate change adaptation issues by communities, inadequate coordination across government levels and sectors, no legal mandate at the local level, no climate change unit at the district and local levels, a lack of knowledge by some staff members tasked with environmental duties at the local level, not enough climate change plans in place at the local level, and outdated information on climate change used in the IDPs. This paper, therefore, recommends that climate change be a standing item in the Integrated Development Plan for local governments, which will ensure that climate change is budgeted for appropriately. In addition, this paper suggests that a mandate for climate change adaptation be developed for all three government levels. There is also a need for the government to invest in capacity development and improve horizontal and vertical coordination to strengthen the weak climate governance capacity that exists.

**Keywords:** climate change adaptation; governance; systems integration; institutional arrangement; barriers; South Africa

## 1. Introduction

Climate change adaptation is defined by the IPCC [1] as the "adjustment in natural or human systems in response to actual or expected climatic stimuli or their effects, which moderates harm or exploits beneficial opportunities". It is estimated that without adaptation, climate change will drive approximately 132 million people into extreme poverty in the next decade alone [2]. Therefore, it is no exaggeration to argue that climate change is a

"wicked" problem [3] or a "super wicked problem" [4]. Climate change is expected to have the largest impact on rainfall, temperature, and water availability in South Africa [5]. The western regions of South Africa are projected to have 30% reduced water availability by 2050 [5]. Van der Bank and Karsten [6] argue that over the last five decades, annual temperatures have increased by approximately 1.5 times in South Africa and 0.65 °C globally. According to the World Economic and Social Survey conducted by Savelli et al. [7] in South Africa, poor and marginalised individuals are likely to experience the worst impacts of future water shortages. This survey supports an earlier study by Nhamo and Agyepong [8], which reported that the overall dam levels in Cape Town, South Africa, dropped from 92.5% to 23% during the 2014–2017 drought, whereas in 2000, severe floods affected northern South Africa, resulting in multiple deaths and damaging infrastructure [9]. In 2016 and 2017, there were reported cases of localised flooding in several provinces across South Africa. Most recently, the 2022 Durban floods had devastating impacts on the lives of many vulnerable people. In light of the above, it is undeniable that the sub-Saharan African region is very susceptible and vulnerable to the impacts of climate change [10]. These impacts have been observed in all sectors of the economy, environment, and society [11]. Given the highly interactive and wicked nature of climate change, many scholars, such as Sibiya et al. [12] and Adom et al. [13], have analysed the impact of climate change on marginalised communities, water, and food security in South Africa. Biesbroek et al. [14] and Keskitalo [15], however, are of the view that the number of studies on climate change adaptation and the forms it takes is far from unlimited, and much remains to be understood about the governance of adaptation and how this may affect its success or failure.

Huitema et al. [16] defined adaptation governance as "the patterns that emerge from the governing activities of social, political, and administrative actors in the realm of climate change adaptation." "Governance is not a set of rules or an activity, but a process; the process of governance is not based on control but on coordination; it involves both the public and private sectors; and it is not a formal institution but a continuing interaction" [17]. Research shows that most of the studies concerning the governance of climate change have focused on the development of the international climate change system, its constituent agreements, the UNFCCC and the Kyoto Protocol, and their implementation [18–21]. While research on barriers to climate change adaptation in South Africa has been conducted previously [22–24], previous research can only be considered a first step towards a more profound understanding of how to overcome these barriers. Furthermore, there is limited empirical evidence in the literature that demonstrates the views of those mandated to drive the climate change adaptation agenda in South Africa. In view of these gaps, an analysis of the barriers to climate change adaptation in South Africa is provided in this paper. The overall aim is to provide an entry point through which the government can develop and enhance a more enabling environment that promotes effective climate change adaptation in South Africa.

The literature review investigates the challenges of climate change adaptation from the global to the local context. In the Section 2 of the paper, the methodology used in this is described. In the Section 3, the empirical evidence of this study is articulated, followed by a comprehensive discussion, conclusions, limitations, and recommendations.

## 2. Literature Review

### 2.1. Theoretical Framework—Barriers to Climate Change Adaptation

The major challenge for successful adaptation is the ability to traverse the barriers that arise in the governance of adaptation [25–28]. Barriers are defined by Adger et al. [29] as the result of action in the realms of finance, culture, and politics that raises questions about the effectiveness and accountability of climate change adaptation. While Moser and Ekstrom [26] define barriers as hurdles that impede adaptation or may necessitate changes that result in missed opportunities or increased costs. They can be overcome, prevented, or minimised by individual or collective action with intensive effort, creative management, changed ways of thinking, political will, and reprioritization of resources, land uses, and

institutions [26]. It is imperative to note that barriers are not viewed the same way by all actors involved in the adaptation process. Thus, Klaus et al. [30] argue that while a barrier may be observed as such by one actor, it may not be the same for another actor' therefore, it depends on how something is valued.

Scholars have used several ways to categorise the barriers to climate change adaptation. The categories used by Adger et al. [29], for example, include institutional, social, informational, financial, and cognitive. On the other hand, Falaleeva et al. [31] use the earth system governance framework, which involves stability, credibility, adaptiveness, and inclusiveness categories. Biesbroek et al. [32] identified seven barrier categories: "conflicting timescales; substantive, strategic, and institutional uncertainty; institutional crowdedness and institutional voids; fragmentation; lack of awareness and communication; motives and willingness to act; and resources". Moser and Ekstrom [26] use a systematic diagnostic framework. The framework uses theories of coupled socioecological systems thinking and multi-level governance theories by focusing on scale, contextual processes, and structures, amongst other factors, and it enables a flexible approach to examining the barriers [33]. Anderies et al. [34] define socioecological systems thinking as "an ecological system intricately linked with and affected by one or more social systems". Berkes [35] views the social (human) and ecological (biophysical) subsystems of systems thinking as two fundamental parts that function as a coupled, symbiotic, and co-evolutionary system. Furthermore, Hooghe and Marks [36] are of the view that theories of multi-level governance can be categorised into two definite types: the first is governance with a distinct structure and a vertically tiered hierarchy, in which only a limited number of authorities have actual decision-making powers [37]. The second type is "polycentric" governance, which is defined by Ostrom [38] as "multiple governing authorities at different scales rather than a monocentric unit".

This systematic diagnostic framework comprises three components, which are identifying the nature of the barrier, its source, and the location of influence over the barrier. In identifying the nature of the barrier, four phases of adaptation, including understanding, planning, implementation, and monitoring, guide this process (as illustrated in Figure 1). According to Moser and Ekstrom [26], the understanding phase focuses on the availability of adequate and valuable information and knowledge and the capacity of actors to engage efficiently with it. The planning phase encompasses the development of adaptation options, the assessment of options, and the selection of options. The management phase focuses on the implementation of the selected options, monitoring the environment and outcome of the realised options, and evaluation.

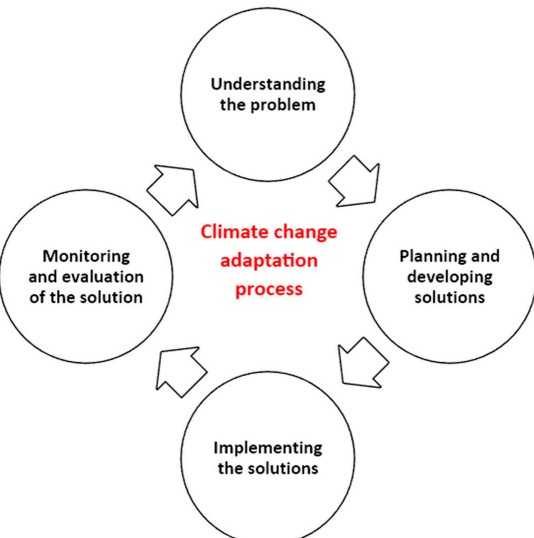

**Figure 1.** Depiction of the systematic diagnostic framework. Source: Reprinted/adapted with permission from Moser and Ekstrom (2010) [26].

The second component of the framework (Figure 2) involves identifying the source. The three essential sources of the barriers include the actors involved in the adaptation process, the larger context in which they act, and the object upon which they act [26,39]. According to Moser and Ekstrom [26], although the system to be managed may produce signs of change, the actors, governance system, and larger human and biophysical context affect whether they are detected and how they are interpreted.

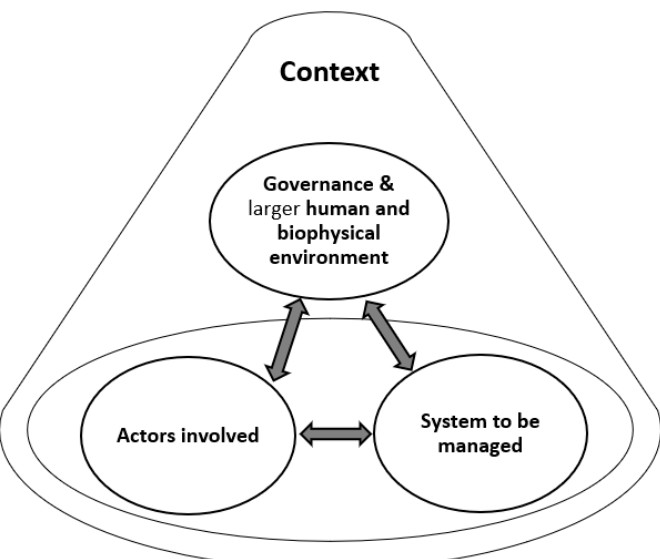

**Figure 2.** The structural elements of the diagnostic framework. Source: Reprinted/adapted with permission from Moser and Ekstrom (2010) [26].

The last component of the systematic diagnostic framework provides a "guide" to identify and understand the barriers, and to design strategies to evade, eradicate, or reduce them. A barrier that is both proximate and contemporary is one over which government officials have direct control here and now. (A in Figure 3), whereas a remote contemporary barrier is one that occurs now but is beyond the official's direct control (B in Figure 3). A proximate legacy barrier is one where a law prevents the official from taking a certain adaptation action (C in Figure 3). A remote legacy barrier is one that is a legacy of past science-policy decisions by remote officials (D in Figure 3).

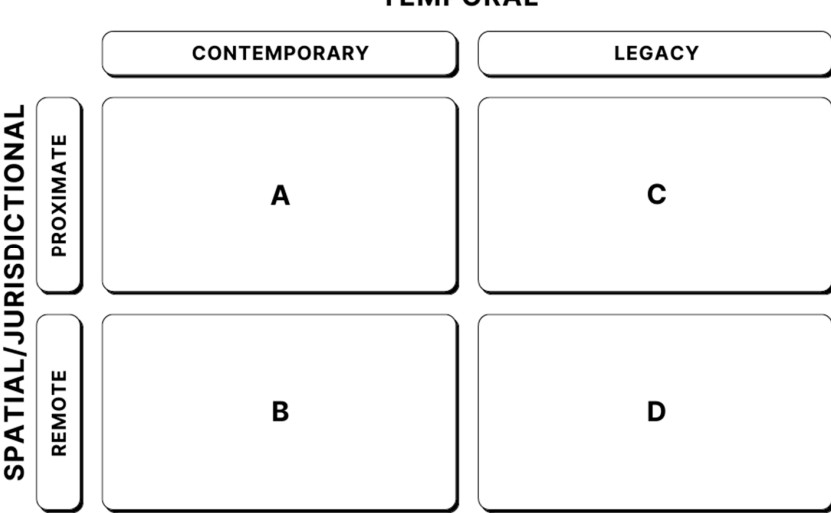

**Figure 3.** Opportunities to overcome barriers. Source: Reprinted/adapted with permission from Moser and Ekstrom (2010) [26].

### 2.2. Barriers to Climate Change Adaptation—A Synthesis of the Literature

Evidence is now emerging that barriers to adaptation often arise from institutional and cognitive constraints [26]. Governance failures occur when there is ineffective institutional decision-making and/or policy implementation [40,41]. These failures limit adaptation by creating barriers and slowing planning and delivery. Ekstrom and Moser [39] identified inadequate financial resources and political support as the common barriers encountered by local people in the San Francisco Bay Area in the United States of America in their adaptation efforts to climate change. This is supported by several studies that argue that at the national, provincial, and municipal government levels, a lack of financial resources and knowledge prevents and "locks out" people and institutions from solving problems and managing change [26,42,43]. Roberts [44], working extensively at the local and international scale on adaptation governance, is of the view that there are limited financial resources available locally to fund climate change adaptation initiatives at the national, provincial, or municipal levels in South Africa.

Furthermore, the problem of capacity has been conclusively cited in the policy integration literature [45]. Cullman et al. [46] argue that the sub-Saharan African region faces persistent technical and financial capacity constraints. Ryan and Bustos [47] argue that government departments and agencies working on climate adaptation policy do not have adequate human and technical abilities to analyse and evaluate the available information. On the other hand, studies conducted in Kenya, Ethiopia, Nigeria, Ghana, and Uganda by Mohmand and Loureiro [48] and Ampaire et al. [49] found that the local governments, though declared self-governing by the central government, do not have sufficient human and financial resources to implement policies or climate-resilient actions, even when adaptation objectives and actions have been integrated into local development plans. A considerable body of literature argues that meeting funding requirements from bilateral and multilateral donors remains difficult for African countries [50,51]. The investment required for adaptation in developing countries can range from USD 50 billion to USD 500 billion annually in 2050 [52]. Numerous studies, therefore, argue that although adaptation finance through the United Nations Framework Convention on Climate Change will help offset some of the climate costs, it is not of the magnitude required for climate-proofing in developing countries [53–55].

The literature also demonstrates that the lack of explicit national agendas and inducements may burden local governments in a different way based on their different capacities [56,57]. Climate adaptation is also viewed as the segregated task of a particular sector that may hamper mainstreaming and horizontal coordination between sectors and departments and coerce the degree to which systems can learn or adapt to climate change [58–63]. It has also been argued that a lack of leadership and accountability at local levels hinders information dissemination, policy implementation, and financial accountability [14,64]. Additionally, it is highlighted in the literature that knowledge gaps are one of the main barriers that affect the planning and implementation of climate change adaptation measures [65]. Ryan and Bustos [47] observed that the knowledge available for climate adaptation policymaking (climate data, impact studies, social-environmental vulnerability assessments, etc.) is dispersed and fragmented. Numerous scholars have also identified limited climate change knowledge, institutional capacity, funding, awareness mechanisms, and coordination amongst and within government institutions as challenges [66,67].

In addition, Füssel [68] and Smit and Pilifosova [69] argue that government institutions lead the process of designing, implementing, and monitoring climate policy, and thus their "political will" is a major factor in the policy success. However, it is widely acknowledged in the literature that it is often a challenge to attain political will. Mossler et al. [70], for example, posit that one of the reasons for the low level of political support for climate change response globally is the way it has been framed by governments. He further argues that framing can shape the perception of a problem and impact the way the audience is likely to address it [70]. In the same vein, Füssel et al. [71] argue that making political decisions based on uncertainty in a dynamic system is a challenge that political leaders are

not always willing or able to face. On the other hand, Smith et al. [72] argue that political will is an important mechanism to overcome bureaucratic resistance and risk aversion in addressing wicked problems like climate change. Clement and Amezaga [73] are of the view that a lack of political will in national policymaking gives only weak inducements to those who implement local policies, which results in a reduced uptake. Keen et al. [74] argue that the political nature of local government means that political interests affect all decisions, including climate adaptation.

The historic tendency of municipalities to view climate change as an environmental problem can also affect mainstreaming, given that environmental issues are assigned a lower priority than other issues [75–77]. Thus, environmental departments are often not assigned much influence or resources [77–79]. Previous studies seem to suggest that politicians have not acknowledged climate adaptation as politically urgent to advance the policy agenda. Subsequently, they identify a tendency to prioritise other political concerns, which are often more tangible issues [58,75,80]. Ziervogel and Parnell [78] argue that one of the challenges that local governments face in mainstreaming adaptation to climate change is the lack of authority held by environmental departments to address climate change. Previous research has blamed the ineffective policy on limited budget allocations and staffing and insufficient stakeholder participation and linkages [81–85]. Therefore, there is strong evidence in the literature of the barriers that hinder the development and implementation of climate change adaptation policies and strategies in developed and developing countries with many pointing to various governance dimensions.

## 3. Materials and Methods

The data collection process for this paper was conducted between February and October 2021. A total of 30 government officials working on climate change adaptation at the national, provincial, and district/local government levels were purposively sampled from the government officials lists received from the Department of Forestry, Fisheries, and the Environment; KwaZulu-Natal Department of Economic Development, Tourism, and Environmental Affairs; and uMkhanyakude District Municipality. Out of the 30 government officials that were sampled, 13 participated in the study, making the response rate 43%. Semi-structured interviews were undertaken online through Microsoft Teams and telephonically with government officials from the Department of Forestry, Fisheries, and the Environment; the KwaZulu-Natal Department of Economic Development, Tourism, and Environmental Affairs; and the uMkhanyakude District Municipality. Government officials provided expert perspectives related to the barriers they are confronted with in facilitating effective climate change adaptation at the three government levels in South Africa. The profiles of the participants are displayed in Table 1 below.

The data obtained from government officials were analysed using thematic analysis. This involved familiarising and engaging with the data, generating codes for the data, searching for themes, reviewing themes, and defining and naming themes. According to Nowell et al. [86], a rigorous thematic analysis can generate insightful and reliable findings. This is aligned with Braun and Clarke [87], who argue that thematic analysis is flexible for identifying, describing, and interpreting themes within a data set in detail. It is tailored to qualitative studies that seek to investigate complex research issues, such as the one this paper sets out to investigate.

This paper used a systematic diagnostic framework developed by Moser and Ekstrom [26] to analyse the barriers to climate change adaptation in South Africa. It was important for a study of this nature to adopt this framework because understanding the barriers to climate change adaptation requires a holistic approach. The systematic diagnostic framework is based on four phases of adaptation, including understanding, planning, implementation, and monitoring. The framework uses theories of coupled socioecological systems thinking and multi-level governance theories by focusing on "scale, contextual processes, and structures, amongst other factors, and it enables a flexible approach to examining barriers" [33]. The systematic diagnostic framework enabled the classification

of the barriers identified in this paper and a comparison of the barriers across various government levels.

**Table 1.** The profiles of the government officials.

| Title | Government Level |
| --- | --- |
| Youth Environmental Coordinator | District/Local |
| District and Local Government support—Environmental Management | District/Local |
| Executive Director—Community Services | District/Local |
| Youth Environmental Coordinator | District/Local |
| District Manager—Environmental Management | Provincial |
| Control Environmental Officer | Provincial |
| Climate Change Intern | Provincial |
| Climate Change Specialist | National |
| Chief Director—Climate Change Adaptation | National |
| Control Environmental Officer | National |
| Director of Adaptive Capacity Programme | National |
| Director—Climate Change Adaptation | National |
| Director—Climate Change Adaptation | National |

Source: Field-based survey notes (2021).

## 4. Results

### 4.1. An Analysis of the Barriers to Climate Change Adaptation in South Africa

The fundamental issue driving this paper is to examine the barriers to climate change adaptation in South Africa. In light of this, government officials working on climate change adaptation were asked, "What are the key challenges that exist in developing and implementing climate change adaptation policies, strategies, and programmes at the national, provincial, and local government levels?" Firstly, government officials who participated in this study identified the barriers to climate change adaptation. Therefore, the nature of the barrier and the frequency of the responses received are presented in Figure 4 (in percentages).

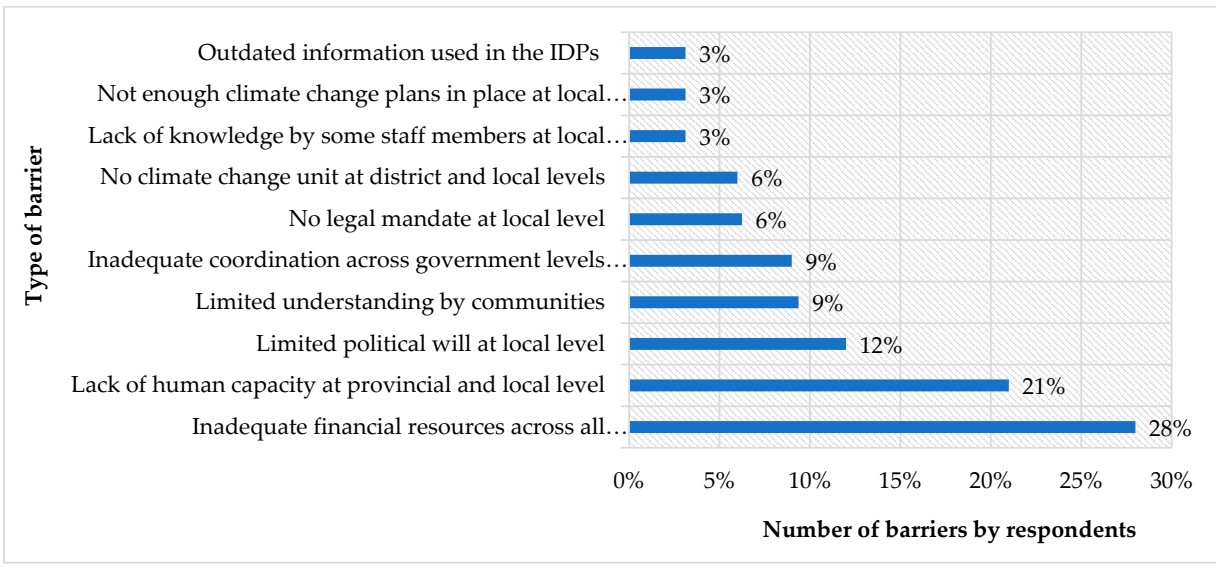

**Figure 4.** Frequency of different types of barriers encountered at the different government levels. Source: Field-based survey notes (2021).

The predominant barrier to climate change adaptation in South Africa is inadequate financial resources across all government levels, which represents 28% of the responses received from government officials working on climate change adaptation (Figure 4). A further 21% indicated that there is a lack of human capacity at the provincial and local government levels. An estimated 12% of the responses claimed that there is limited political will at the local government level, whereas 9% expressed that there is ineffective coordination across government officials, and a further 9% indicated there is a limited understanding of climate change issues by the communities. Of the total 12% of responses, 6% indicated that there is no climate change unit at the district and local government levels, and the remaining 6% expressed that there is no legal mandate at the local government level. Of the total 9% of responses, 3% claimed that outdated information on climate change is being used in the municipal Integrated Development Plans (IDPs); 3% expressed that there are not enough climate change plans in place at the local government level; and another 3% indicated a lack of knowledge by some staff members who are tasked with environmental duties at the district and local government levels.

In addition, the fundamental sources of the barriers were identified by government officials working on climate change adaptation issues. This data is presented using the systematic diagnostic framework; the data presented in Table 2 is structured around identifying the fundamental sources of the barriers under the four phases of adaptation (i.e., understanding, planning, implementation, and monitoring guide the process).

**Table 2.** A summary of the barriers presented by phase.

| Understanding | Planning | Implementation and Monitoring |
|---|---|---|
| Limited understanding of climate change issues by communities | Not enough climate change plans in place at the local level | Inadequate financial resources |
| Limited political will | | Lack of human resources |
| Lack of climate change knowledge by some staff members tasked with environmental duties | Inadequate coordination across government levels and sectors | |
| | No climate change unit at the local level | |
| | No legal mandate at the local level | |
| Outdated information on climate change issues used in the IDPs | | |

Source: Field-based survey notes (2021).

An examination of Table 2 suggests that four of the barriers have to do with understanding the importance of climate change adaptation. These include limited understanding of climate change issues by communities; lack of climate change knowledge by some staff members tasked with environmental duties; outdated information on climate change issues used in the municipal Integrated Development Plans; and limited political will. In view of this, one of the government officials indicated that:

> *"The change in political parties or administration after elections is always problematic because different political parties have different priorities and oftentimes climate change is not prioritised. Thus, it is important to note that politics is a reality we must learn to navigate"* (Remark 1).

Furthermore, Table 2 demonstrates that there are four barriers to climate change adaptation in the planning phase, one of which is that there are not enough climate change plans in place at the local level, and the other three overlap between the planning and implementation and monitoring phases (i.e., no legal mandate at the local level, no climate change unit at the local level, and inadequate coordination across government levels and sectors). To validate the above assertions, one government official alluded to the following:

> *"There isn't a united voice across government levels. It is not because people are in denial, but people are dealing with real-life issues, e.g., human settlements and service delivery.*

*However, we need to work together. We need to find people within the different sectors to take up climate change work and work together using a bottom-up approach. We need to be proactive rather than reactive. We need to continue working at the operational level so that it cascades up"* (Remark 2).

Lastly, inadequate financial resources and a lack of human resources are the main barriers encountered in the implementation and monitoring phase. A government official from the Department of Forestry, Fisheries, and the Environment, who participated in an interview, disclosed that:

*"There are not enough funds to implement all climate change plans and projects"* (Remark 3).

On the other hand, another government official indicated that:

*"Although we receive external funding from donors, it is never enough"* (Remark 4).

In view of the lack of human resources, one government official stated that:

*"The lack of employees who are purely climate change practitioners at the local government level hinders progress"* (Remark 5).

Social-ecological systems are nested within a larger system at global, regional, national, and local scales. Thus, it is important to note that the above-mentioned barriers vary across the government scales. In view of this, using the second component of the systematic diagnostic framework, the location of the influence over the barrier is identified in Table 3.

**Table 3.** A comparison of the barriers across the various government levels.

| Barriers | National Level DFFE | Provincial Level KZN EDTEA | District/Local Level UKDM |
|---|:---:|:---:|:---:|
| Inadequate financial resources | X | X | X |
| Lack of human resources | | X | X |
| Limited political will | | | X |
| Limited understanding of climate change issues by communities | | | X |
| Inadequate coordination | X | X | X |
| No legal mandate | | | X |
| No climate change unit | | | X |
| Lack of climate change knowledge by some staff members tasked with environmental duties | | | X |
| Not enough climate change plans | | | X |
| Outdated information on climate change issues used in the IDPs | | | X |

Source: Field-based survey notes (2021).

The information in Table 3 suggests that there are inadequate financial resources and inadequate coordination across all three government levels in South Africa. Of particular interest is the fact that Table 3 seems to suggest that all the barriers (i.e., inadequate financial resources, lack of human resources, limited political will, limited understanding of climate change issues by communities, inadequate coordination, no legal mandate, no climate change unit, lack of climate change knowledge by some staff members tasked with environmental duties, not enough climate change plans, and outdated information on climate change issues used in the IDPs) are encountered or felt at the local government level in South Africa.

*4.2. Suggested Strategies to Overcome Barriers*

It was imperative to try and identify some of the origins of the barriers that were identified in this paper. Therefore, using the last component of the systematic diagnostic framework, Figure 5 demonstrates the origins of the barriers and how this has an influence on overcoming the barriers to climate change adaptation in South Africa.

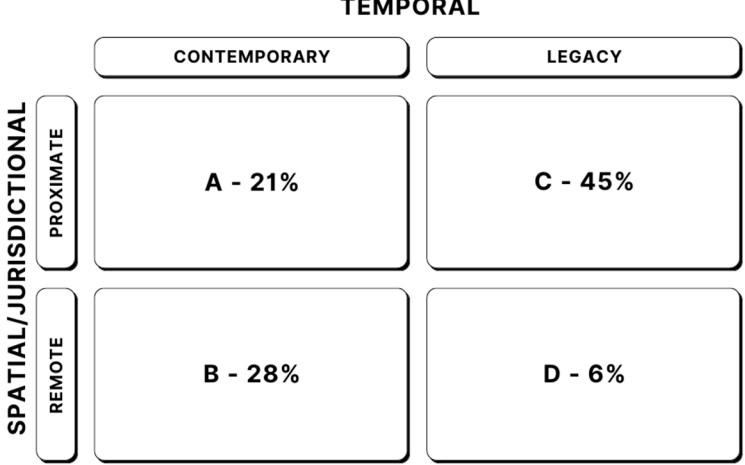

**Figure 5.** Summary of the origins of the barriers and the contribution represented as percentages. Source: Reprinted/adapted with permission from Moser and Ekstrom (2010) [26] using field-based survey notes (2021).

Scrutinising Figure 5 suggests that a combined total of 21% of the responses on the existing barriers form part of the proximate contemporary barriers. This comprises limited understanding of climate change issues by communities, representing 9% of the total responses, a lack of climate change knowledge by some staff members tasked with environmental duties (3%), and inadequate coordination across government levels and sectors (9%). These barriers can be overcome by government officials working on climate change adaptation, as they have control over them. The strategies that can be used to overcome these barriers include, but are not limited to, the following:

- Creating awareness and educating communities on climate change issues;
- Improving coordination across the government levels and sectors;
- Ensuring that staff members that are tasked with climate change issues are well-versed in this area of expertise.

A further 28% of the total responses form part of the remote contemporary barriers, which, although they occur now, the government officials working on climate change adaptation have no control over (i.e., the issue of inadequate financial resources). Therefore, the government can do the following:

- Prioritise climate change adaptation issues to ensure that they are budgeted for appropriately;
- Source external funding from donors for all government levels in order to ensure that there are funds to implement the strategies and plans that are meant to facilitate climate change adaptation on the ground.

A total of 45% of the total responses form part of the proximate legacy barriers against which the law prevents action. This comprises no legal mandate at the local levels (6%), no climate change unit at the district and local levels (6%), lack of human capacity at the provincial and local levels (21%), and limited political will at the local levels (12%). However, government officials have control over initiating changes in the regulations by:

- Advocating for the need for the government to create a legal mandate at the local government level;
- Creating a climate change unit at the local government level;

- Increasing the human capacity to ensure that there are climate change adaptation practitioners on the ground implementing the strategies and plans that have been developed. In addition, these practitioners can play a major role in engaging with the communities, educating the communities, and driving change on the ground.

While the remaining 6% of the barriers form part of the "remote legacy barriers", which are a legacy of past decisions (i.e., not enough climate change plans in place at the local government level; representing 3% and the use of outdated information in the IDPs of municipalities; representing 3% of the total responses), some of these barriers could still be addressed by:

- Ensuring that the IDPs have relevant, up-to-date information on environmental issues, which can be achieved by having individuals who are well-versed in this area of expertise contribute to such important municipal documents and discussions.

However, closer scrutiny of Figure 5 suggests that the proximate barriers provide a combined 66% of the barriers over which government officials working on climate change adaptation have control by either providing the action to overcome the barriers or initiating changes in the regulations that exist. In view of this assertion, government officials shared the following suggestions:

- *"There is room for creating more climate change adaptation awareness programmes"* (Remark 6).
- *"Climate change adaptation strategies, plans and programmes must be aligned across the three levels of government and across sectors which are affected by climate change"* (Remark 7).
- *"There is an opportunity for improved integration between the three levels of government when it comes to climate change governance"* (Remark 8).

Therefore, the overall impression represented in Figure 5 indicates that although there are barriers that the law prevents, the government officials working on climate change adaptation should initiate and drive change.

## 5. Discussion

The findings presented in this paper demonstrate that the two predominant barriers to climate change adaptation in South Africa are inadequate financial resources, which are cross-cutting across all three government levels, and a lack of human capacity at the provincial and local levels. These barriers are mostly encountered in the implementation and monitoring phases of adaptation. The results presented in this study are consistent with previous findings from other studies. Ngwenya and Simatele [42] and Averchenkova et al. [88], for example, argued that at the national, provincial, and municipal government levels, a lack of financial resources and knowledge inhibits people and institutions from resolving problems and managing change. South Africa's National Treasury [89] estimates that nearly 80% of the district's municipal revenue is from the national government, and because of competing socio-economic demands, little of this revenue is allocated to the climate change response. Additionally, the empirical evidence presented in this paper supports previous studies that argue that financial barriers limit municipal adaptation mainstreaming, partly due to the many responsibilities they must perform and due to their lack of institutional autonomy [75,78]. These findings align with the findings from other regions. Measham et al. [75], for example, found that the key challenge to climate change adaptation across the Sydney region is the constrained resources (financial and human) faced by local governments. Hooli [90] found that in Namibia, the government has limited financial support for local or informal adaptation responses. As observed by Pressend [91], there is often an inadequate commitment to international funding for climate change adaptation actions, and when this funding is available, it is difficult to access it due to excessive red tape.

Goebel [92] concurred that a human resources crisis exists within South African municipalities, which must be addressed to overcome policy implementation hurdles. Similarly, the South African National Development Plan 2030, "Our Future, Make It Work",

considers that there is a lack of human resources and skills at the local government level. Alemaw and Simatele [23] further argue that the lack of resources at all government levels is inclined to inhibit people and institutions from solving problems and mapping changes caused by climate change. Having established this, we argue that the South African government must "put their money where their mouth is" and invest strategically in climate change to build a transformative climate change governance system. In addition, it is important to have a thought process on how South Africa will best utilise climate change adaptation finance for the benefit of all. Achieving this requires a radical shift towards enhancing investments in climate change funding in South Africa for adaptation.

Furthermore, the findings demonstrate that there is limited political will at the local government level, and this is because of poor understanding of climate change issues by politicians. This finding aligns with those of several previous studies. CoGTA [93], for example, acknowledged that a lack of policy coherence, skill scarcity, a lack of political leadership, and corruption are some of the major barriers encountered in promoting effective climate change adaptation in South Africa. This is supported by Khambule [94], who argues that weakening institutional ability and arrangements within municipalities takes place through political interference, a lack of efficient bureaucracy, and gross corruption. On the other hand, Scoville-Simonds et al. [95] are of the view that there are three major political adaptation difficulties, "related to differential responsibility, the global uneven production of vulnerability, and unequal relations of power in adaptation decision-making". Mapfumo et al. [96] suggest that political will and political probability are important factors in undertaking structured measures that are transformational in response to climate change. There is a need for South African political representatives to demonstrate political will for climate change adaptation when they make their national, provincial, and district/local budgets. Although there are conflicting priorities, such as service delivery through water provision, housing, energy supply, and infrastructure, climate change adaptation must be featured at the top of the list, particularly because climate change has and will continue to have dire impacts on all subsectors of the economy.

In addition, the findings presented in this paper revealed that three of the barriers to climate change adaptation alluded to by government officials had to do with understanding climate change issues. This involved a limited understanding of climate change adaptation issues by communities, a lack of knowledge by some staff members tasked with environmental duties at the local level, and outdated information used by politicians in the municipal IDPs. In view of this, we argue that climate change adaptation requires shared understanding. This finding largely corroborates Ross et al. [97], who argue that efficient and collective adaptation to climate change requires shared understandings of the possible impacts and risks gained through informed presentation and discussion with other participants. Numerous scholars, including Lemos et al. [98], argue that relations between knowledge producers and those who could use it can often overcome the cognitive and social divisions that weaken information usability by building trust, improving awareness, and improving understanding around issues of accuracy and reliability of climate information [99–102]. In view of this, therefore, this study argues that the government of South Africa must educate all citizens on climate change adaptation in order to improve understanding, increase awareness, and build trust.

It was also revealed that there is inadequate coordination across all government levels and that this affects the planning, implementation, and monitoring phases of adaptation. This paper suggests that there is a need for government officials working on climate change adaptation issues to advocate for coordination across government levels and sectors, as this will allow for resource mobilisation to address the wicked problem we are faced with. Loring et al. [103] argue that well-operated collaboration and coordination from each level of governance based on partnerships can facilitate adaptation to climate change and make societies more resilient to the uncertainty of impacts posed by climate change stakeholders. Some researchers also believe that learning through coordination improves local actors' capacity to adapt, enhances policy-making opportunities in broader governance networks,

and provides training for policymaking [104,105]. This is also supported by Mpandeli et al. [106], who argue that coordination ensures that there is resource mobilisation and policy convergence across sectors.

The findings of this study also demonstrate that two of the barriers to climate change adaptation alluded to by government officials have a negative impact on the planning, implementation, and monitoring phases of the adaptation process. These barriers include the lack of a legal mandate at the local government level and the fact that there is no climate change unit at the district and local levels. This finding seems to align with Du Plessis and Kotzé [107], who are of the view that the absence of a climate change mandate in South African municipalities has not helped matters as it has "downplayed the seriousness of the need for climate governance and action," making municipal authorities reluctant to act. Furthermore, this study argues that it is time for South Africa to tackle the long-standing barriers and move towards a governance system that will build the resilience and adaptive capacity of societies in times of changing climatic conditions.

It is suggestive that one barrier arises due to poor planning because there are not enough climate change plans at the local level. Without proper adaptation planning and the necessary investments in social safety nets, climate risks can adversely affect efforts to achieve the UN Sustainable Development Goals (SDGs) in sub-Saharan Africa [108–110].

The findings presented in this paper demonstrate that there is a differential distribution of resources across the different government levels. Therefore, it can be argued that the coordination among the multilevel polycentric governance structures and agency can become feasible and robust if coupled and symbiotic relationship is established as proposed by the social systems approach. Although multilevel governance systems may create more challenges and obstacles in the coordination among multilevel polycentric governance structures and decision making, social systems theory will enable us to overcome such challenges. This is in line with Biermann et al. [111] and Ostrom and Janssen [112], who argue that although multilevel governance theory enables learning and succeeding at multiple scales, there is no guarantee that it can successfully deal with complex human–ecological systems. Furthermore, it is congruent with numerous scholars who are of the view that to cope with the "wicked" problem of climate change, innovative and theoretically controversial adaptation actions whose goal is to support the associated social-ecological adaptation are necessary [113–115].

In view of the above discussion, it is important to note that the situation is not all doom and gloom. As illustrated in Figure 5, the majority of the barriers that exist are barriers over which government officials working on climate change adaptation have control by either providing actions to overcome the barriers or initiating changes in the regulations that exist. Thus, this paper argues that government officials working on climate change adaptation in South Africa should drive change, as they are better positioned to advocate for improvement in the conditions in which they are expected to drive the climate change adaptation agenda of the country.

## 6. Conclusions

This paper identifies some of the barriers to climate change adaptation in South Africa. These barriers include inadequate financial resources, a lack of human capacity at the provincial and local levels, limited political will at the local level, limited understanding of climate change adaptation issues by communities, inadequate coordination across government levels and sectors, no legal mandate at the local level, no climate change unit at the district and local levels, a lack of knowledge by some staff members tasked with environmental duties at the local level, not enough climate change plans in place at the local level, and outdated information on climate change used in the IDPs. There is, therefore, a need to improve capacity development, strengthen policy alignment, and improve integration between the three levels of government.

In addition, it is important for local municipalities to strengthen their role in issues related to climate change. Thus, we recommend that climate change be mainstreamed

in the Integrated Development Plan for local governments, which will ensure that when projects are budgeted for, they are climate responsive. Motivation from a climate change-responsive Integrated Development Plan will also ensure that there are funds to hire more climate change adaptation practitioners. Furthermore, there must be effective coordination of climate change adaptation issues across government levels and sectors if we are to effectively manage climate change.

*Limitations and the Way Forward*

Finally, this research is subject to a few limitations. Firstly, this study focused only on the barriers related to climate change adaptation. While our results reaffirm the importance of investigating and addressing the barriers to climate change adaptation, we are aware that barriers are highly context-specific, challenging to compare, and difficult to use for a more generalised understanding. Thus, there are opportunities to bolster our findings through future research. Most notably, our analysis focuses narrowly on barriers to climate change adaptation without taking into consideration the barriers encountered by government officials working in the subsectors of the economy that are affected by climate change (e.g., water, agriculture, and energy). We recommend that these limitations be addressed and explored in future studies.

**Author Contributions:** Conceptualization, N.P.S. and M.D.S.; methodology, N.P.S.; validation, R.K.A. and M.D.S.; formal analysis, N.P.S.; investigation, N.P.S.; writing—original draft preparation, N.P.S.; writing—review and editing, N.P.S., D.K.D., C.V., S.P.M., L.Z., M.A.K., M.S., R.K.A. and M.D.S.; supervision, M.D.S.; funding acquisition, M.D.S.; resources, M.D.S. All authors have read and agreed to the published version of the manuscript.

**Funding:** This research was funded by the National Research Foundation of South Africa, Research Grant No. 129481, Ref RCUZ200513521731, and the University of the Witwatersrand Postgraduate PhD Merit Award.

**Institutional Review Board Statement:** The study was reviewed and approved by the University of the Witwatersrand Ethics Committee (protocol number H20/10/28, date of approval: 16 October 2020).

**Informed Consent Statement:** Written informed consent has been obtained from the participants to publish this paper.

**Data Availability Statement:** The original contributions presented in the study are included in the article; further inquiries can be directed to the corresponding authors.

**Acknowledgments:** Thank you to all the research participants from uMkhanyakude District Municipality, KwaZulu-Natal Department of Economic Development and Environmental Affairs, and the Department of Forestry, Fisheries, and the Environment for their invaluable insights.

**Conflicts of Interest:** The authors declare no conflict of interest. The funders had no role in the design of the study; in the collection, analysis, or interpretation of data; in the writing of the manuscript; or in the decision to publish the results.

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
