# Peer review of "Overcoming Bureaucratic Resistance: An Analysis of Barriers to Climate Change Adaptation in South Africa"

_climate, doi:10.3390/cli11070145_

Round 1

Reviewer 1 Report (Previous Reviewer 1)

The authors have adequately addressed my concerns from the first review. I am happy to recommend for publication. 

Author Response

Reply to Reviewer 1

There were no comments received from Reviewer 1.

Reviewer 2 Report (New Reviewer)

The study addresses an important area and research is urgently needed in order to understand how government can be more effective in addressing climate change.

However, the paper has some weaknesses.

Firstly, it is solely based on government official's perceptions. It is unclear who these officials are? What is their role, level, level of education? Junior officials in particular departments may have a very different view to senior officials responsible for programmes. It is therefore important to provide this information. It would also be useful to understand these perceptions against the perceptions of others or to test them against a framework. Currently, the study reflects perceptions of people which may, or may not, be an accurate representation of reality. 

Secondly, a number of concepts such as climate change adaptation and governance are poorly defined. It would have been useful to define these clearly and develop a suitable theoretical framework that could be used to establish what was ádequate ie in terms of policy, capacity, finance etc. This could then be used as the basis of the evaluation. There are widely used climate adaptation frameworks in South Africa, such as the Green Book, which are not mentioned in the study. These could contribute to a theoretical framework. 

Thirdly, no survey questions are provided. Without this we cannot ascertain how questions have been asked and whether this may have influenced answers. Similarly, responses are not analysed in detail and results appear anecdotal.

Fourth, a study of this nature should be able to provide detailed recommendations that can be used by government to improve their effectiveness. The results and findings in their current form are not specific and do not make a significant contribution.

Address details above.

Please ensure that Materials and Methods include a detailed methodology instead of describing the area. 

Conclusions

Please review conclusions of the paper against the research questions, the methodology and the data gathered through the study.

Does the study establish that there is a lack of human resources? By how much? Of what type? Where? Does the study establish there is lack of financial resources? By how much? What are expected benchmarks? Does the study establish a lack of political will? How? 

Care should be taken that the study does not over claim and conclusions are restricted to findings actually achieved in the study. Conclusions should be specific.

This paper established the capacity and institutional challenges to the governance of climate change adaptation in South Africa. These challenges include a lack of human capacity, financial resources, and a lack of political will, fragmented sectoral planning, and organizational silos.

Author Response

Reply to Reviewer 2

Point 1: Firstly, it is solely based on government official's perceptions. It is unclear who these officials are? What is their role, level, level of education? Junior officials in particular departments may have a very different view to senior officials responsible for programmes. It is therefore important to provide this information. It would also be useful to understand these perceptions against the perceptions of others or to test them against a framework. Currently, the study reflects perceptions of people which may, or may not, be an accurate representation of reality.

 Response 1: Thank you very much for pointing this out. We have included the profiles of the government officials that participated in the study on which this paper is based. Furthermore, we have used a framework to analyse the barriers to climate change adaptation in South Africa.

Point 2: Secondly, a number of concepts such as climate change adaptation and governance are poorly defined. It would have been useful to define these clearly and develop a suitable theoretical framework that could be used to establish what was adequate i.e. in terms of policy, capacity, finance etc. This could then be used as the basis of the evaluation. There are widely used climate adaptation frameworks in South Africa, such as the Green Book, which are not mentioned in the study. These could contribute to a theoretical framework.

Response 2: Thank you very much for the comment. I have defined the concepts such as climate change adaptation (see L40-42) and governance (see L70-75). Kindly see pages 3–5 for the theoretical framework adopted. Although the Green Book was suggested by the reviewer, it appears that it is more of a risk assessment tool than a tool to assess barriers. Therefore, we felt that the systematic diagnostic framework was the most relevant framework to adopt for this paper.

Point 3: Thirdly, no survey questions are provided. Without this we cannot ascertain how questions have been asked and whether this may have influenced answers. Similarly, responses are not analysed in detail and results appear anecdotal.

Response 3: Thank you very much for the comment. The research question that was asked has been included in the paper. We have analysed the data using thematic analysis and data is presented along the lines of diagnostic framework that was adopted in this paper.

Point 4: Fourth, a study of this nature should be able to provide detailed recommendations that can be used by government to improve their effectiveness. The results and findings in their current form are not specific and do not make a significant contribution.

Response 4: Thank you very much for the comment. We have tried to improve this.

Point 5: Please ensure that Materials and Methods include a detailed methodology instead of describing the area.

Response 5: Thank you very much for pointing this out. We have tried to improve this section. Kindly see L253-288. 

Point 6: Please review conclusions of the paper against the research questions, the methodology and the data gathered through the study.

Response 6: Thank you very much for pointing this out. We have tried to improve this section. Kindly see 573-591.

This manuscript is a resubmission of an earlier submission. The following is a list of the peer review reports and author responses from that submission.

Round 1

Reviewer 1 Report

Overcoming bureaucratic resistance: Capacity and institutional 2 challenges to a transformative climate governance system in 3 South Africa 4 Nomfundo Sibiya 1*, Mulala Danny Simatele 1, 2 and Richard Kwame Adom

Abstract: This paper investigates into the climate governance system in South Africa and examines the challenges encountered by the actors involved in implementing climate change adaptation actions. It provides an abstract outlining the methods, problems, findings and recommendations. South Africa is one of the fastest growing economies which is facing the conflicting challenges of economic growth and climate change actions. The paper is a timely reminder of the struggles of many developing countries that are at the forefront of climate change.

Introduction: The introduction provides some background to the climate change situation in South Africa with reference to relevant studies. The first sentence refers to IPCC third assessment report 2001. I would have expected a relevant reference to the most recent report (IPCC sixth assessment report 2022). The section briefly identifies research gap in institutional analysis and outlines the sequence of different sections.

Literature review: There is no denying the fact that institutions play an important role in implementing government actions in climate change and enforcing the legal mandate to environmental governance. There is a complex nexus between the actors implementing climate actions. These relationships complicate how each actor functions in developing countries. One universal example that can be used here is local political leadership and its influence on the local institutions. The literature review focuses on the global context of climate governance identifying the various factors that are working as hindrances to institutional efficiencies. It starts with studies from outside of South Africa. The references seem adequate and help identify some of the important institutional challenges in climate change adaptation. The literature review then delves into what is happening in South Africa in terms of climate change governance at three levels of government, namely, national, provincial and local/municipal. Using reference to relevant literature, it identifies several barriers to effective environmental governance at each level. The challenges are not new. They seem to appear in all recent literature on climate governance in developing countries.

What I find lacking in the literature review is that the authors have failed to assert the literature gap that justifies the current research. Using the sentence “Therefore, there is a need to address these barriers in order to have working institutions in South Africa which are committed to facilitating effective and transformative climate change governance.”, is not enough to say that a study similar to previous studies on South Africa is warranted. The authors are in best position to say why this study was necessary in the first place and I would like to see more assertive statements along this vein.

Materials and methods:

The study area: Unless there is more information on the study area later in the article, the authors need to provide further descriptions. Also, the purpose of generalization the authors have used a sentence (lines 248-250) “The study area is representative of many rural regions in KwaZulu-Natal that continue to grapple with poverty and the impacts of climate change on their livelihood-supporting systems.” My question is ‘How?’. I do not think this sentence is adequate to justify the selection of the study area as representative of the rest of the region. Overall, this section is weak and the authors need to add further to this section. Use of direct quotations of interviews and their purpose need to be outlined briefly in the methods section.

Data Collection and Data Analysis: A survey was conducted to gather data using Google Forms online. The authors used a random sampling technique to select key stakeholders. The key stakeholders, as described in the paper, are chiefs, leaders and officials from different wakes of life and government and non-governmental organizations. I think the authors should tell readers more about the selection criteria and process of key stakeholders. The authors used descriptive statistics to analyse quantitative data and identified themes to interpret qualitative data. Identification of themes from qualitative data is supposed to be a rigorous process. Readers would be interested to know how it was done and whether any software package was used.

Results: The fist sentence of this section is “One of the objectives of this paper is to assess perceptions on the current climate change adaption governance system in South Africa”. This objective has not been outlined in the abstract or in the introduction. ‘Perception’ and ‘perspective’ are two different words with different meanings. The authors need to make this section consistent with the abstract and introduction sections. Table1 on page 8 of 21 needs to have a number and a title on the table.

Discussion: This is a good section. It summarizes the findings for the study supported by findings from other recent studies in the region. There is scope for further refinement in how this section aligns with the abstract and the introduction sections. The authors need to make sure that what is claimed in the abstract and the introduction sections have been proven through the methods, results and discussions sections. Pages 452-453 “To achieve this, climate change adaptation policies and strategies must be translated into the different languages indigenous to South Africa.” Has this been identified by the participants as an impediment to effective implementation of climate change policies? If yes, this needs to be highlighted in the findings sections with quotations and arguments.

The authors have identified knowledge gap through discussions and survey but do not say much about community participation and local knowledge system and how it can contribute to knowledge and understanding of climate change adaptation, limitations and opportunities.

This is timely research for South Africa, the country which is facing the effects of climate change like many other developing countries. The authors have effectively employed established methods of data collection, analysis and interpretation. Use of mixed method has provided methodological rigour and authentication. The findings are consistent with the findings of similar studies elsewhere. Studies in climate change are complex as they incorporate diverse aspects and phases of the phenomenon. They include pure scientific studies in the physical nature of climate change namely change in temperature, precipitation etc. On the other spectrum of the scale, they investigate socio-economic and socio-political aspects. Again, in this area of conflict and confusion, researchers try to understand another aspect which is called ‘climate governance’, that is, how climate change policies and projects are formulated and implemented and what to examine if one wants to understand its effectiveness. In this arena authors have examined the challenges of climate change governance.

Overall, this article has all the elements of a good publication, but it needs further refinement along the line of recommendations made by this reviewer. The paper is not well-knit from the beginning to the end. There needs to be consistency between all sections so that they flow through and read well. The research gap is not well-identified and the importance of the research is not well-established.

Reviewer 2 Report

none